**Data Availability Statement:** "All data are publicly available at https://www.cdc.gov/nchs/nhanes/index.htm". Others can access this data in the same way that authors use the information in our

# Association between serum Klotho and the prevalence of osteoarthritis: A cross-sectional study from NHANES 2007–2016

Yue Qiu[1][☯], Huangyi Yin[2][☯], Jinzhi Meng[1], Yang Cai[1], Junpu Huang[1], Xifan Zheng[1], Jun Yao[1], Jia Li [3]*

1 Osteoarticular Surgery, The First Affiliated Hospital of Guangxi Medical University, Nanning, China,
2 Geriatric Department of Endocrinology, The First Affiliated Hospital of Guangxi Medical University, Nanning, China, 3 Department of Pathology, The First Affiliated Hospital of Guangxi Medical University, Nanning, China

☯ These authors contributed equally to this work.

\* jialee2005@126.com

## Abstract

### Background

Osteoarthritis (OA) is a degenerative joint disease prevalent in the elderly. Currently, the relationship between the senescence inhibitor Klotho and OA remains unclear. This study investigated the relationship between serum soluble Klotho (S-Klotho) and OA.

### Methods

This cross-sectional study was based on the 2007–2016 National Health and Nutrition Examination Survey (NHANES). Three multifactorial logistic regression models were constructed to assess the association between serum Klotho and OA. Restricted cubic spline (RCS) curves were further used to assess whether there was a nonlinear relationship between serum Klotho and OA. Finally, stratified analyses and interaction tests were used to evaluate the association's stability. To further investigate the relationship between serum Klotho and OA, we recruited 107 patients for analysis at the First Affiliated Hospital of Guangxi Medical University.

### Results

The final 8,918 participants included in this study comprised 50.55% females and 49.45% males, with 18.10% of participants suffering from OA and a mean S-Klotho level of 846.41 (5.61) pg/ml. All three logistic regression models observed a negative association between continuous S-Klotho and OA risk. When S-Klotho was categorized into tertiles, the fully adjusted model showed that participants in the third tertile had a 17% lower risk of OA than those in the first tertile (OR = 0.83, 95% CI: 0.70, 0.99, $P$ = 0.035). The RCS curves showed a linear negative association between S-Klotho and the incidence of OA ($P$ for overall = 0.025; $P$ for non-linearity = 0.667). Further subgroup analyses and interaction tests suggested that the negative association between S-Klotho and OA remained stable in different

Data Availability statement. And the author does not have special access to the data that anyone else does not have.

**Funding:** This project was supported by the Nanning Qingxiu District Science and Technology Plan Project (grant/award number: 2020018), Guangxi Medical and Health Appropriate Technology Development and Extension and Application Project (grant/award number: GZSY22-62), Guangxi Science and Technology Base and Talent Special Project (grant/award number: GuikeAD19254003) and Health Department of Guangxi Zhuang Autonomous Region Self-funded project (grant/award number: Z2013039).

**Competing interests:** The authors have declared that no competing interests exist.

**Abbreviations:** OA, osteoarthritis; CVD, cardiovascular disease; S-Klotho, soluble Klotho; SNPs, single-nucleotide polymorphisms; NHANES, National Health and Nutrition Examination Survey; NCHS, National Center for Health Statistics; ELISA, Enzyme-Linked Immunosorbent Assay; OD, optical density; PIR, poverty to income ratio; BMI, body mass index; PA, physical activity; MET, metabolic equivalents; DM, diabetes mellitus; CKD, chronic kidney disease; RCS, restricted cubic spline.

conditions. Research conducted in China has shown that the negative correlation between serum Klotho levels and the prevalence of OA remains evident among Chinese individuals (OR: 0.77, 95% CI: 0.66, 0.90, P<0.001).

## Conclusion

Our study suggests that elevated levels of the senescence inhibitor S-Klotho may be a potential protective factor for OA, which may provide new insights into the diagnosis and treatment of OA.

## Introduction

Osteoarthritis (OA) is the most common degenerative joint disease affecting the elderly; the main manifestations are chronic swelling, pain and limited movement of the joints [1]. The prevalence of OA has been rising yearly due to the increasing aging population and the obesity epidemic in older adults worldwide [2]. During 1999–2017, the age-standardized prevalence of OA was reported to have increased by 9.3% globally, with the United States being one of the countries with the most significant increase, rising by 23.2% [3]. By 2030, the number of Americans with OA will reach 67 million, which will grow to 78 million by 2040 [4, 5]. Studies have shown that OA is not only a potential risk factor for cardiovascular disease (CVD) mortality [6, 7], but also one of the most common causes of disability in older adults [8], posing a severe economic and public health threat. However, current treatments for OA are limited, and severe cases require arthroplasty. Therefore, early identification and development of new effective non-surgical therapies is imperative.

In 1997, Makoto Kuro-o et al. found that mice with Klotho gene mutations exhibited a shortened lifespan and a range of aging symptoms [9]. Since then, the Klotho gene has attracted the attention of many researchers as a potential anti-aging target [10]. The Klotho gene is mainly expressed in brain and kidney tissues, and encodes three Klotho proteins: α-Klotho, β-Klotho, and γ-Klotho [11]. Among them, α-Klotho has received extensive attention as its extracellular structure can be hydrolyzed to soluble Klotho (S-Klotho) to enter the circulation and act on other organs to exert hormone-like effects [10, 12]. S-Klotho has been reported to be involved in pathological processes, such as oxidative stress, apoptosis, Inflammation, and cellular senescence [13–16]. For middle-aged and older adults over 40, S-Klotho levels gradually decrease with age [17]. In addition, clinical studies have reported that decreased S-Klotho levels are associated with the occurrence of a range of aging-related diseases such as cognitive impairment, frailty, osteoporosis, and increased mortality [18–20]. Therefore, Klotho may be a potential target for preventing aging-related diseases and life extension.

The mechanisms underlying the development of OA are not completely clear, but it is generally recognized that the etiology of OA may involve age, gender, genetics, mechanical injury, and obesity [21]. Among them, aging is considered the most critical risk factor for the development of OA [22]. Whether Klotho, an anti-aging gene, is involved in the pathogenesis of OA has attracted the attention of researchers. The Klotho gene has been reported to regulate calcium and phosphorus metabolism and the pathogenesis of osteoporosis and articular subchondral sclerosis [18, 23, 24]. In vivo studies have revealed a significant decrease in Klotho expression levels in the articular cartilage of OA mice, while Klotho over-expression inhibited chondrocyte damage [25–28]. Cartilage aging is mainly manifested as extracellular matrix

sclerosis [28]. Recent studies have found that the aging extracellular matrix impairs chondrocyte integrity by promoting methylation of the α-Klotho gene, ultimately leading to OA [29]. This is regarded as the main reason for the occurrence of age-associated OA. In addition, studies by Zhang F et al. and Tsezou A et al. have demonstrated that single-nucleotide polymorphisms (SNPs) in the human Klotho gene are associated with susceptibility to OA, but these studies were limited to Greeks and Britons [30, 31].

The hormone-like effect of S-Klotho is one of the crucial pathways by which α-Klotho proteins exert their anti-aging effects. However, no study has reported a relationship between serum S-Klotho and the prevalence of OA. Based on previous studies, we propose the scientific hypothesis that serum S-Klotho may be negatively associated with the incidence of OA. Based on the National Health and Nutrition Examination Survey (NHANES), the present study investigates the relationship between serum S-Klotho and the incidence of OA in middle-aged and older adults.

## Methods

### Data source

NHANES is an extensive, continuous cross-sectional survey sponsored by the National Center for Health Statistics (NCHS). The study collects basic information and dietary patterns, as well as a detailed physical examination and blood examinations to assess the nutritional and health status of participants. This study was approved by the NCHS Ethics Review Committee, and written informed consent was obtained from all participants.

To further validate the relationship between serum Klotho levels and OA, we conducted a study at the Orthopedic and Joint Surgery Department of the First Affiliated Hospital of Guangxi Medical University. This study was approved by the Ethics Committee of the First Affiliated Hospital of Guangxi Medical University (Approval Number: 2024-E566-01), and written informed consent was obtained from all participants.

### Study populations

Since NHANES only investigated S-Klotho levels in 13,764 participants aged 40–79 years from 2007–2016, the present study was conducted on these participants. As shown in Fig 1, this study sequentially excluded 1,602 participants who lacked diagnostic information on OA, 1,541 subjects with a diagnosis of other arthritis, and 1,703 participants who were missing covariables. Finally, 8,918 eligible participants were included in this study.

Between August 6, 2024, and September 13, 2024, a total of 107 patients were recruited at the First Affiliated Hospital of Guangxi Medical University. The cohort comprised 51 patients with OA and 56 patients without OA. Participants who sought treatment for sports injuries and were excluded from an OA diagnosis were classified as the non-OA group.

### Diagnosis and evaluation of osteoarthritis

OA was diagnosed based on the results of questionnaires administered by professionals. Participants were considered to meet the diagnostic criteria for OA if they self-reported having been told by a doctor or professional that they had arthritis and the diagnostic type of arthritis was OA. Previous studies have confirmed the consistency of self-reported OA with clinically diagnosed OA [32].

In the Chinese study, the diagnosis of OA was based on the criteria set forth in the 2018 Chinese Osteoarthritis Diagnosis and Treatment Guidelines: 1) Recurrent pain within the past month; 2) Morning stiffness lasting ≤30 minutes; 3) Age ≥50 years; 4) Crepitus during

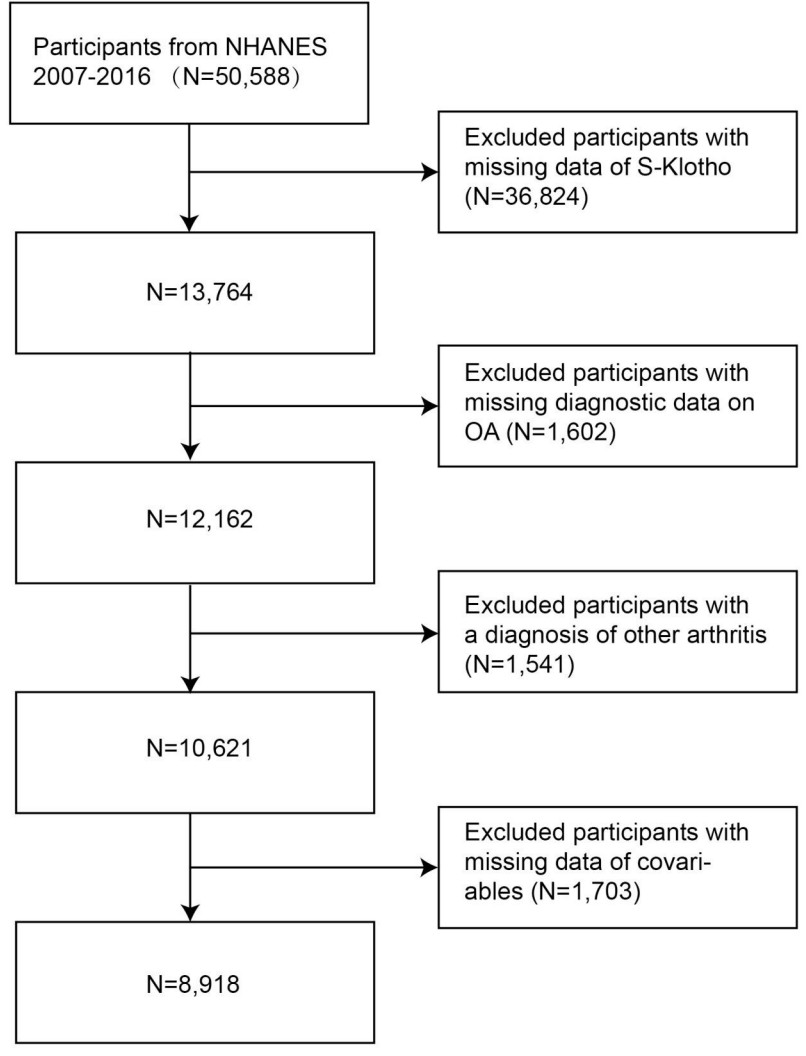

**Fig 1. Flow chart of the participants exclusion process.**

movement; 5) X-ray findings showing joint space narrowing, subchondral bone sclerosis or cyst formation, and osteophyte formation at the joint margins. OA can be diagnosed if the first criterion is met along with any two of the remaining criteria [33].

### Serum soluble Klotho measurements

Serum samples from all participants were stored at -80°C and assayed by an Enzyme-Linked Immunosorbent Assay (ELISA) kit manufactured by IBL International Corporation, Japan, with a practical sensitivity of 4.33 pg/ml. Each sample was assayed twice, and the average of the two measurements was typically used as the final S-Klotho concentration. If the difference between the two results is greater than 10%, retesting is required. The final S-Klotho concentration data for our study were obtained from the official NHANES website.

In the Chinese study, serum samples from all patients were analyzed using the Klotho protein ELISA kit (FANKEW, F9713-B). A standard curve was established by measuring the

optical density (OD) values of the standards provided with the kit. The concentration of each sample was determined based on the standard curve.

## Covariables

Based on previous studies, the present study considered possible confounders. These included demographic characteristics, lifestyle habits, and comorbidities. Demographic characteristics included gender (female, male), age (40–50 years, 50–60 years, 60–70 years, > 70 years), race (non-Hispanic black, non-Hispanic white, Mexican American, other), marital status (single, non-single), poverty to income ratio (PIR) (> 1.3%, 1.3–3.5%, ≥ 3.5%), education level (less than high school, high school graduate, more than high school), and body mass index (BMI) (< 25kg/m$^2$, 25-30kg/m$^2$, ≥ 30kg/m$^2$). Lifestyle habits included smoking status, drinking status, and physical activity (PA). Specifically, smoking status was categorized as never smokers, previous smokers, and current smokers based on whether or not they had smoked more than 100 cigarettes in their lifetime and whether or not they had smoked in the last year. Alcohol consumption was categorized into never drinkers, former drinkers, and current drinkers based on whether or not they had more than 12 drinks in their lifetime and whether or not they had consumed alcohol in the past year. Current drinkers were further categorized into three groups: light drinkers (≤ 1 drink/day for women, ≤ 2 drinks/day for men, or ≤ 1 day of heavy drinking per month); moderate drinkers (≤ 2 drinks/day for women, ≤ 3 drinks/day for men, or 2–5 days of heavy drinking per month); and heavy drinkers (≥ 3 drinks per day for women, ≥ 4 drinks per day for men, or ≥ 5 days of heavy drinking per month). Weekly PA needed to be measured based on metabolic equivalents (MET) and subdivided into four categories: inactive (0 MET-min/week), mildly active (< 600 MET-min/week), moderately active (600–1200 MET-min/week), and heavily active (> 1200 MET-min/week).

Comorbidities included diabetes mellitus (DM), hypertension, chronic kidney disease (CKD), CVD, and hyperlipidemia. DM was defined as self-reported DM or taking hypoglycemic medication. Participants were diagnosed with hypertension by meeting more than one of the following conditions: 1) hypertension that a physician had reported; 2) taking blood pressure-lowering agents; 3) an elevated mean of blood pressure (mean systolic blood pressure over 140 mmHg or diastolic blood pressure over 90 mmHg). Hyperlipidemia was defined as meeting more than one of the following conditions: 1) triglycerides ≥ 150 mg/dl; 2) LDL-C ≥ 130 mg/dl; 3) total cholesterol ≥ 200 mg/dl; 4) HDL-C < 50 mg/dl for female or HDL-C < 40 mg/dl for male; 5) taking lipid-lowering medications. CKD was defined as persistent proteinuria ≥ 30 mg/g or eGFR < 60 (ml/min/1.73m$^2$). Participants self-reported congestive heart failure, coronary heart disease, heart attack, stroke, and angina diagnosed as CVD.

## Statistical analysis

This study statistically analyzed participants using appropriate weights to represent the health status of noninstitutionalized residents of the United States. Continuous and categorical variables were expressed as means (standard errors) and sample sizes (percentages), respectively. Subjects were categorized into two groups based on the occurrence of OA, and baseline characteristics were compared between the two groups. Continuous variables were compared using ANOVA or the Kruskal-Wallis rank test, while categorical variables were compared using the chi-square test. Considering the high values of S-Klotho, we statistically analyzed the S-Klotho of all participants after logarithmic transformation. Then, three multifactor logistic regression models were constructed to further explore the association between serum S-Klotho and OA. Model 1 was not adjusted for any variables. Model 2 was adjusted for gender, race, and age. Model 3 was further adjusted for marital status, education level, PIR, smoking status,

alcohol consumption, PA, BMI, DM, hypertension, hyperlipidemia, CVD, and CKD based on model 2. In addition, Restricted cubic spline (RCS) curves were plotted to visualize the association between serum S-Klotho and OA in this study. Finally, stratified analyses and interaction tests based on gender, age, race, hypertension, DM, and BMI were performed to assess the stability of this association further.

In the study conducted in the First Affiliated Hospital of Guangxi Medical University, an independent two-sample t-test was used to compare Klotho levels between OA and non-OA patients. Subsequently, logistic regression analysis was employed to explore the potential relationship between serum Klotho levels and the prevalence of OA.

The study was statistically analyzed using R software (version 4.3.2), and $P<0.05$ was defined as statistically significant.

## Results

### Comparison of baseline characteristics of participants

There was a total of 8,918 participants in this study, of which 49.45% were males. The average serum S-Klotho level of all participants was 846.41 (5.61) pg/ml, of which the participants with OA accounted for 18.10% of the total population. Participants were divided into two groups for comparison based on the occurrence of OA. Table 1 showed that participants with OA had greater age, BMI, and lower levels of S-Klotho and were more likely to be female compared to participants without OA. Additionally, the comparison of race, education level, smoking status, drinking status, and prevalence of comorbidities (hypertension, DM, CKD, CVD, and hyperlipidemia) between the two groups was also statistically significant.

### Association between serum S-Klotho and osteoarthritis

The association between serum S-Klotho and OA was explored by log-transforming serum S-Klotho for all participants and grouping them by tertiles of log-transformed S-Klotho levels. When S-Klotho was analyzed as a continuous variable, the results of all three multifactorial logistic regressions showed that higher serum S-Klotho levels were related to a lower prevalence of OA. In model 3, participants in the third tertile of S-Klotho had a 17% lower risk of OA development than participants in the first tertile (OR = 0.83, 95% CI: 0.70, 0.99, $P$ = 0.035) (Table 2). In addition, we further visualized the negative association between serum S-Klotho and OA using RCS curves. After adjusting for all confounders, the results showed a significant linear negative relationship between them ($P$ for overall = 0.025; $P$ for non-linearity = 0.667) (Fig 2).

### Stratification analysis

To further investigate the stability of the negative association between serum S-Klotho and OA, stratified analyses and interaction tests were conducted in this study. The results showed that the negative association between serum S-Klotho and OA remained stable when stratified by gender, age, race, BMI, hypertension, and DM (Fig 3).

### Validation of the relationship between serum Klotho and the prevalence of osteoarthritis

To further validate the inverse relationship between serum Klotho levels and the prevalence of OA, we collected blood samples from 107 patients at the Department of Orthopedics, First Affiliated Hospital of Guangxi Medical University. This study included 51 OA patients and 56 non-OA individuals. Compared to non-OA patients, OA patients exhibited significantly

**Table 1. Comparison of weighted baseline characteristics of participants according to the occurrence of osteoarthritis.**

| Variables | Total N = 8918 | Non-osteoarthritis N = 7304 | Osteoarthritis N = 1614 | P-value |
|---|---|---|---|---|
| Sex (%) | | | | < 0.001 |
| Female | 4393(50.55) | 3374(47.19) | 1019(63.68) | |
| Male | 4525(49.45) | 3930(52.81) | 595(36.32) | |
| Age (%) | | | | < 0.001 |
| < 50 years | 2763(33.90) | 2575(39.51) | 188(12.00) | |
| 50–60 years | 2410(31.59) | 2048(32.46) | 362(28.20) | |
| 60–70 years | 2338(22.56) | 1736(18.93) | 602(36.73) | |
| ≥ 70 years | 1407(11.95) | 945(9.10) | 462(23.06) | |
| Race (%) | | | | < 0.001 |
| Non-Hispanic Black | 1627(8.09) | 1371(8.66) | 256(5.86) | |
| Non-Hispanic White | 4038(74.98) | 3053(72.55) | 985(84.51) | |
| Mexican American | 1397(6.37) | 1240(7.24) | 157(2.95) | |
| Other | 1856(10.56) | 1640(11.56) | 216(6.68) | |
| Marital status (%) | | | | 0.253 |
| No | 2968(28.23) | 2381(27.80) | 587(29.91) | |
| Yes | 5950(71.77) | 4923(72.20) | 1027(70.09) | |
| Education level (%) | | | | 0.003 |
| Below high school | 2217(14.16) | 1906(14.85) | 311(11.46) | |
| High school graduate | 1909(21.20) | 1568(21.66) | 341(19.41) | |
| College or above | 4792(64.64) | 3830(63.49) | 962(69.13) | |
| PIR (%) | | | | 0.350 |
| < 1.3 | 2453(15.45) | 2032(15.66) | 421(14.61) | |
| 1.3–3.5 | 3207(31.99) | 2617(31.58) | 590(33.61) | |
| ≥ 3.5 | 3258(52.56) | 2655(52.76) | 603(51.78) | |
| Smoking (%) | | | | < 0.001 |
| Never | 4693(53.16) | 3950(55.05) | 743(45.77) | |
| Former | 2544(29.38) | 1945(26.90) | 599(39.06) | |
| Now | 1681(17.46) | 1409(18.05) | 272(15.17) | |
| Drinking (%) | | | | < 0.001 |
| Never | 1232(9.95) | 1022(10.13) | 210(9.25) | |
| Former | 1783(16.34) | 1394(15.46) | 389(19.78) | |
| Mild | 3199(40.71) | 2575(39.98) | 624(43.57) | |
| Moderate | 1276(17.05) | 1044(17.14) | 232(16.71) | |
| Heavy | 1428(15.95) | 1269(17.30) | 159(10.69) | |
| PA (%) | | | | < 0.001 |
| Inactive | 2427(22.70) | 1884(21.17) | 543(28.66) | |
| Low | 1303(14.48) | 1047(14.26) | 256(15.37) | |
| Median | 1009(11.52) | 842(11.46) | 167(11.76) | |
| High | 4179(51.30) | 3531(53.11) | 648(44.21) | |
| BMI (%) | | | | < 0.001 |
| <25 kg/m$^2$ | 2228(25.69) | 1930(27.13) | 298(20.08) | |
| 25–30 kg/m$^2$ | 3204(36.39) | 2733(37.88) | 471(30.57) | |
| ≥ 30 kg/m$^2$ | 3486(37.92) | 2641(34.99) | 845(49.35) | |
| DM (%) | | | | < 0.001 |
| No | 7404(87.24) | 6145(88.47) | 1259(82.40) | |
| Yes | 1514(12.76) | 1159(11.53) | 355(17.60) | |

*(Continued)*

**Table 1.** (Continued）

| Variables | Total<br>N = 8918 | Non-osteoarthritis<br>N = 7304 | Osteoarthritis<br>N = 1614 | P-value |
|---|---|---|---|---|
| Hypertension (%) | | | | < 0.001 |
| No | 4442(54.60) | 3888(58.43) | 554(39.61) | |
| Yes | 4476(45.40) | 3416(41.57) | 1060(60.39) | |
| CVD (%) | | | | < 0.001 |
| No | 7931(90.82) | 6602(92.33) | 1329(84.92) | |
| Yes | 987(9.18) | 702(7.67) | 285(15.08) | |
| Hyperlipidemia (%) | | | | < 0.001 |
| No | 1846(20.85) | 1583(21.95) | 263(16.56) | |
| Yes | 7072(79.15) | 5721(78.05) | 1351(83.44) | |
| CKD (%) | | | | < 0.001 |
| No | 7273(85.39) | 6054(87.01) | 1219(79.06) | |
| Yes | 1645(14.61) | 1250(12.99) | 395(20.94) | |
| S-Klotho (pg/ml) | 846.41(5.61) | 853.27(6.15) | 819.59(8.06) | < 0.001 |

Continuous variables were expressed as mean (standard error) and categorical variables were expressed as frequencies (percentages).

PIR: poverty income ratio; PA: physical activity; BMI: body mass index; DM: diabetes mellitus; CVD: cardiovascular disease; CKD: chronic kidney disease; S-Klotho: soluble Klotho.

higher serum Klotho levels (P<0.001) (S1 Table). After adjusted race, sex, age, increased serum Klotho levels were associated with a reduced prevalence of OA, with an odds ratio (OR) of 0.77 (OR: 0.77, 95% CI: 0.66, 0.90, P<0.001) (S2 Table).

## Discussion

As far as we know, this is the first study to explore the association of serum S-Klotho with OA in Americans. The results of this study show that participants with OA tended to be older and have lower levels of S-Klotho. Elevated serum S-Klotho was associated with a reduced risk of OA, with a significant linear negative relationship. Study in China have also confirmed the negative correlation between serum Klotho levels and the prevalence of OA. Gender, age, race, BMI, hypertension, and DM did not influence the negative relationship between serum S-Klotho and OA.

**Table 2. Association between serum S-Klotho and osteoarthritis.**

| Exposures | Model1<br>[OR (95% CI) P-value] | Model2<br>[OR (95% CI) P-value] | Model3<br>[OR (95% CI) P-value] |
|---|---|---|---|
| Log S-Klotho (continuous) | 0.77(0.68,0.88) <0.001 | 0.83(0.72,0.95) 0.010 | 0.84(0.73,0.97)0.021 |
| Log S-Klotho (tertiles) | | | |
| Q1 | ref | ref | ref |
| Q2 | 0.85(0.72,1.01)0.063 | 0.91(0.75,1.09) 0.295 | 0.92(0.76,1.11)0.383 |
| Q3 | 0.77(0.66,0.89) <0.001 | 0.82(0.70,0.97)0.021 | 0.83(0.70,0.99)0.035 |
| P for trend | <0.001 | 0.022 | 0.036 |

Model 1 was not adjusted for any variables. Model 2 was adjusted for gender, race, and age. Model 3 was adjusted for gender, race, age, marital status, education level, PIR, smoking status, alcohol consumption, PA, BMI, DM, hypertension, hyperlipidemia, CVD, and CKD.

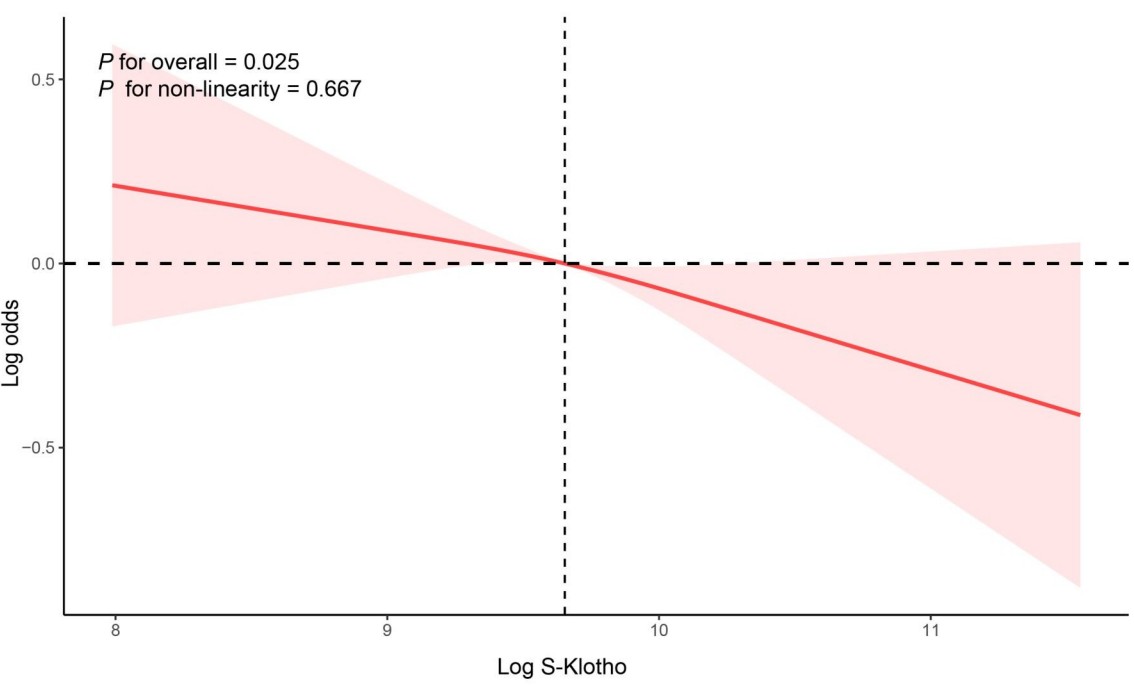

**Fig 2. Restricted cubic spline fitting for the association between S-Klotho and osteoarthritis.**

OA is the most common degenerative joint disease affecting the elderly, and long-term joint pain and activity limitation leads to decreased quality of life for the elderly, placing a heavy socioeconomic burden on the community [1]. Although the pathogenesis of OA has not

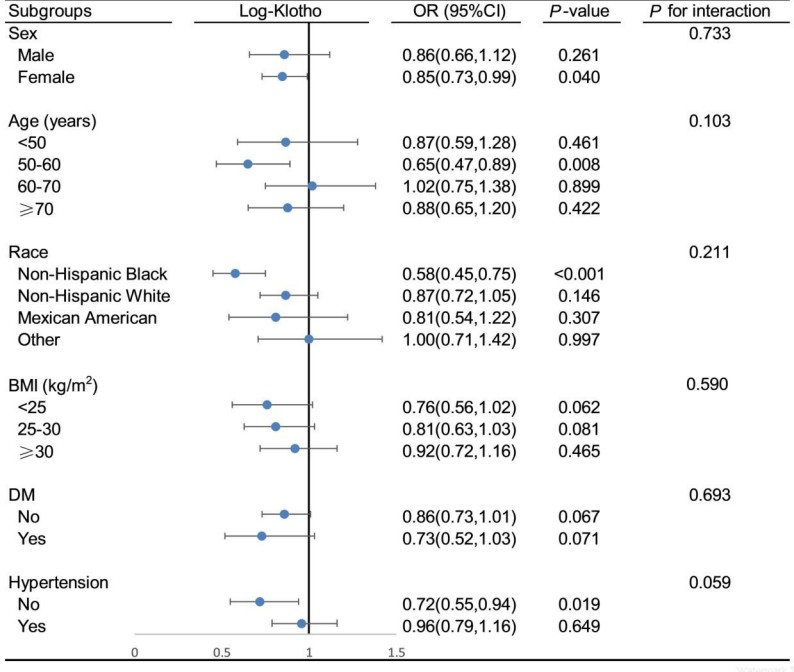

**Fig 3. Subgroup analysis and interaction test of the association between S-Klotho and osteoarthritis.**

been fully elucidated, it is currently considered that the pathogenesis of OA may involve aging, gender, genetics, diet, mechanical injury, and obesity [21]. Of these, aging is thought to possibly have the greatest impact on the development of OA [22]. The prevalence of OA increases with age. The Framingham study showed that the prevalence of OA in people aged 60–70 years was 33%, while it rose to 43.7% in the population over 80 years of age [34]. Another cohort study revealed that the prevalence of symptomatic hip OA was as high as 17% in subjects aged 75 years or older, while the prevalence in subjects aged 45–54 years was only 5.9% [35]. Studies have revealed that inflammatory factor levels such as IL-1, IL-6 and TNF-α gradually increase with age, leading to chronic low-grade inflammation throughout the body [36, 37]. Age-related systemic or localized inflammation is an important reason for the development of OA [2]. In addition, the aging of cartilage is thought to be associated with defective autophagy, manifested explicitly in the decreased ability of the body to process dysfunctional organelles and proteins [38–40]. The expression of crucial proteins for autophagy declines progressively with age [41]. In vivo studies have also found that humans or mice with OA have reduced expression of essential proteins of autophagy compared to the normal group [42]. Interestingly, age-related increases in oxidative stress appear to contribute to the development of OA by modulating chondrocyte apoptosis [43].

Since 1997, the Klotho gene has been recognized as a potential target for extending lifespan and suppressing age-related diseases [9]. Klotho mutant mice exhibit a range of aging symptoms such as osteoporosis, atherosclerosis, and skin atrophy, accompanied by a shortened lifespan [9, 44]. Klotho genes are involved in the regulation of renal tubular calcium and phosphate reabsorption, which affects bone formation [45]. Based on this, researchers have conducted further clinical studies investigating the relationship between Klotho gene expression and the risk of developing OA in humans. A survey in Caucasian women showed that four functional genetic variants in the Klotho gene were associated with the risk of hand OA [30]. Another study also found that two SNPs (G395A and C2998T) in the Klotho gene were associated with susceptibility to knee OA in the Greek population [31]. In the presence of lytic enzymes, the extracellular structure of α-Klotho protein is cleaved to S-Klotho to enter the humoral circulation and act on target organs to exert anti-aging effects [10]. In addition, Klotho has shown promise in predicting mortality in arthritis populations. A cohort study of patients with rheumatoid arthritis found that elevated Klotho levels within specific ranges (838.81 pg/mL for all-cause mortality and 1061.23 pg/mL for cardiovascular mortality) were associated with a reduced risk of both all-cause and cardiovascular mortality [46]. The relationship between S-Klotho and the risk of OA has not been elucidated. Therefore, the present study focused on the association between S-Klotho and OA. Consistent with previous studies, our study found that elevated levels of S-Klotho were associated with a reduced risk of developing OA in middle-aged and older adults, and this relationship remained consistent in most cases. Additionally, the small-scale study conducted in China further validated the negative correlation between serum Klotho levels and OA.

Furthermore, several basic studies have demonstrated that the Klotho gene may have a potentially beneficial effect in inhibiting articular cartilage damage and the development of OA. In vivo studies have shown that articular chondrocytes from OA mice have lower Klotho gene expression than normal chondrocytes [25–28]. When human OA chondrocytes were cultured with secreted α-Klotho, the expression of OA-associated cartilage damage factors, NOS2 and MMP13, was suppressed [26]. The occurrence of OA is associated with cartilage damage-induced oxidative stress [47], and Klotho is considered to be a potent inhibitor of oxidative stress [14, 48]. It has been shown that Klotho can protect chondrocytes from OA-induced oxidative stress via the NLRP3/caspase-1/IL-1β axis and the Trx/Prx antioxidant system [27]. Matrix metalloproteinases (MMP) are essential enzymes that regulate the degradation of the

articular cartilage matrix and are modulated by the Wnt signaling pathway [49–52]. The latter is an essential pathway for the regulation of bone and cartilage homeostasis and is one of the major causes of OA [53, 54]. Studies have shown that the down-regulated Klotho gene in vivo acts on Wnt family molecules in OA models, leading to activation of the Wnt signaling pathway, which further up-regulates MMP and induces chondrocyte damage [28]. A recent study demonstrated that age-associated extracellular matrix stiffness, by initiating the mechanical signaling pathway, further leads to methylation of the α-Klotho gene promoter, decreasing α-Klotho expression and contributing to chondrocyte damage, ultimately leading to the development of OA [29]. In addition, intra-articular injection of adeno-associated virus (AAV) particles carrying soluble α-Klotho proteins could help to inhibit the progression of OA and promote cartilage repair [55].

## Strengths and limitations

Based on the extensive study of NHANES, this study pioneered the exploration of the association between serum S-Klotho and OA, providing further theoretical support for developing the Klotho gene as a potential target for the diagnosis and therapy of OA. This study was weighted for all participants, and the results are applicable to the entire population of non-institutionalized American residents. However, we must recognize some shortcomings of this study. Importantly, we also conducted a small-scale study in the First Affiliated Hospital of Guangxi Medical University to further validate the relationship between serum Klotho levels and OA. First, this was a cross-sectional study and a causal relationship between the S-Klotho and the prevalence of OA could not be inferred. Second, this study may still have some potential confounders that affect the relationship between serum S-Klotho and OA. In addition, the diagnosis of OA in this study was dependent on the self-reported medical history of the participants, which may be subject to recall bias. Finally, all participants in this study were Americans, and the conclusion may not be appropriate for generalization to other countries. Despite the limited sample size and potential information gaps in the validation study, we have undeniably established a preliminary relationship between serum Klotho levels and OA. This provides a theoretical foundation for future large-scale, prospective cohort studies across various regions and populations to further explore the relationship between serum Klotho levels and OA.

## Conclusion

In conclusion, serum S-Klotho is linearly and negatively associated with the prevalence of OA in middle-aged and older Americans. This study supports that the anti-aging factor Klotho may be a potential target for the diagnosis and treatment of OA.

## Supporting information

**S1 Table. Baseline characteristics of the Chinese population.**
(DOCX)

**S2 Table. Association between serum Klotho and osteoarthritis in the Chinese population.**
(DOCX)

**S1 File. Data summary of osteoarthritis in China.**
(XLSX)

**S2 File. Summary of Nhanes data 2007–2016.**
(TXT)

## Acknowledgments

We are grateful to all the staffs and participants in the NHANES.

## Author Contributions

**Conceptualization:** Jia Li.

**Data curation:** Yue Qiu, Huangyi Yin, Junpu Huang.

**Formal analysis:** Yue Qiu.

**Investigation:** Huangyi Yin, Yang Cai, Junpu Huang.

**Methodology:** Huangyi Yin.

**Project administration:** Jun Yao.

**Resources:** Jun Yao.

**Supervision:** Jia Li.

**Validation:** Jinzhi Meng.

**Visualization:** Yue Qiu, Jinzhi Meng, Xifan Zheng.

**Writing – original draft:** Yue Qiu.

**Writing – review & editing:** Jia Li.

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
