## [Decision Letter · Decision Letter 0]

31 Jul 2024

PONE-D-24-19425Association between serum Klotho and the prevalence of osteoarthritis: a cross-sectional study from NHANES 2007-2016PLOS ONE

Dear Dr. Li,

Thank you for submitting your manuscript to PLOS ONE. After careful consideration, we feel that it has merit but does not fully meet PLOS ONE’s publication criteria as it currently stands. Therefore, we invite you to submit a revised version of the manuscript that addresses the points raised during the review process.

We look forward to receiving your revised manuscript.

Kind regards,

Calogero Caruso, MD

Academic Editor

PLOS ONE

Journal Requirements:

"This project was supported by the Nanning Qingxiu District Science and Technology Plan Project (grant/award number: 2020018), Guangxi Medical and Health Appropriate Technology Development and Extension and Application Project (grant/award number: GZSY22-62), Guangxi Science and Technology Base and Talent Special Project (grant/award number: GuikeAD19254003) and Health Department of Guangxi Zhuang Autonomous Region Self-funded project (grant/award number: Z2013039)."

Reviewers' comments:

Reviewer's Responses to Questions

**Comments to the Author**

1. Is the manuscript technically sound, and do the data support the conclusions?

Reviewer #1: Yes

2. Has the statistical analysis been performed appropriately and rigorously? 

Reviewer #1: Yes

3. Have the authors made all data underlying the findings in their manuscript fully available?

Reviewer #1: Yes

4. Is the manuscript presented in an intelligible fashion and written in standard English?

Reviewer #1: Yes

5. Review Comments to the Author

Reviewer #1: Paper is well designed and data are clearly reported however in my opinion some points should be clarified :

Authors using data from NHANES database have found an association among low levels of s-klotho and the patients self reported osteoarthritis. Even if self reported OA might be considered appropriate to categorize the patient groups the possibility of a bias (as authors stated too)in the classification of the patients groups.

By a methodological point of view Authors should clarify if they analyzed data stored in National Center for Health Statistics database or received samples on which they measured s-klotho

I think that is inappropriate define “races” instead of populations (table 1) Is very particular the classification in Non-Hispanic Black, Non-Hispanic White, Mexican American and other. Why authors do not used classical Caucasian, black, Hispanic and Asiatic definition?

Regarding ethics authors should clarify if participants in NHANES whose data were analyzed gave consent for their samples to be used in future research

I agree, as reported by the authors that “the conclusion may not be appropriate for generalization to other countries” than USA. In this view I would like to suggest to the authors to recruit a little cohort of OA patients (>100<300 individuals) on which evaluate the negative correlation between s-klotho and OA presence

Discussion on the biological bases of klotho levels influencing OA presence might be enriched discussing data in the light of the following paper: Che QC, Jia Q, Zhang XY, Sun SN, Zhang XJ, Shu Q. A prospective study of the association between serum klotho and mortality among adults with rheumatoid arthritis in the USA. Arthritis Res Ther. 2023;25(1):149. doi:10.1186/s13075-023-03137-0

6. PLOS authors have the option to publish the peer review history of their article (what does this mean?). If published, this will include your full peer review and any attached files.

Reviewer #1: No

---

## [Author Response · Author response to Decision Letter 0]

30 Sep 2024

Dear Editors and Reviewers: 

Thank you for the editors/reviewers’ comments concerning our manuscript entitled 

“Association between serum Klotho and the prevalence of osteoarthritis: a cross-sectional study from NHANES 2007-2016” (EMID:bdb7c4b0b766ad59). Thank you again for your positive comments and valuable suggestions to improve the quality of our manuscript. We have studied comments carefully and have made correction which we hope meet with approval. Modifications are highlighted in different colors in the article. The main corrections in the paper and the responds to the editors/reviewers’ comments are as following:

Journal Requirements:

Response：

We sincerely thank you for carefully reading. We have revised the manuscript according to the PLOS ONE style templates.

"This project was supported by the Nanning Qingxiu District Science and Technology Plan Project (grant/award number: 2020018), Guangxi Medical and Health Appropriate Technology Development and Extension and Application Project (grant/award number: GZSY22-62), Guangxi Science and Technology Base and Talent Special Project (grant/award number: GuikeAD19254003) and Health Department of Guangxi Zhuang Autonomous Region Self-funded project (grant/award number: Z2013039)."

If this statement is not correct you must amend it as needed. Please include this amended Role of Funder statement in your cover letter; we will change the online submission form on your behalf.

Response (this part of the revision is highlighted in Yellow)：

Thanks for your careful review. In the Author Contributions section of the original manuscript, we have mentioned the contribution of the funder Yao J, as described in the manuscript: “Qiu Y, Meng JZ, Yao J, Cai Y, Huang JP, and Zheng XF involved in preparing and visualization of the results”. To clarify this description, we will revise it to: “Yao J (the funder), Qiu Y, Meng JZ, Cai Y, Huang JP, and Zheng XF involved in the preparation of the original manuscript and the visualization of the results”. (line 408-410)

Response (this part of the revision is highlighted in Blue)：

Thank you for your valuable suggestion. In the Methods section of the original manuscript, we mentioned the full name of the IRB or ethics committee and whether informed written or verbal consent was obtained: “The study was formally conducted after approval by the NCHS Ethics Review Committee and obtaining informed consent from all participants”. To enhance clarity and meet the journal's high standards, we will revise this statement to: “This study was approved by the NCHS Ethics Review Committee, and written informed consent was obtained from all participants”. The original ethical approval documents have been uploaded as supplementary materials. (line 110-111)

Reviewer Comments: 

Reviewer 1

1. Authors using data from NHANES database have found an association among low levels of s-klotho and the patients self reported osteoarthritis. Even if self reported OA might be considered appropriate to categorize the patient groups the possibility of a bias (as authors stated too)in the classification of the patients groups.

Response：

Thank you for your careful review. According to previous NHANES studies, the diagnosis of osteoarthritis is based on questionnaire surveys conducted by medical professionals, which suggests that self-reported osteoarthritis possesses a certain degree of accuracy and research value. Consistent with the diagnostic methods used in our study, Li Chen et al. investigated the relationship between urinary heavy metals and osteoarthritis[1], Donghui Zhao et al. examined the relationship between flavan-3-ol monomer intake and osteoarthritis[2], Jie Huang et al. explored the link between lipid accumulation products and osteoarthritis[3], Yuxuan Liu et al. examined the association between omega-3 polyunsaturated fatty acids and osteoarthritis[4], and Hongfei Xue et al. assessed the relationship between visceral fat metabolism scores and osteoarthritis risk[5]. All of these studies used self-reported osteoarthritis data for diagnosis.

References：

[1]Chen L, Zhao Y, Liu F, et al. Biological aging mediates the associations between urinary metals and osteoarthritis among U.S. adults. BMC Med. 2022;20(1):207. Published 2022 Jun 17. doi:10.1186/s12916-022-02403-3

[2]Zhao D, Shen S, Guo Y, et al. Flavan-3-ol monomers intake is associated with osteoarthritis risk in Americans over 40 years of age: results from the National Health and Nutritional Examination Survey database. Food Funct. 2024;15(13):6966-6974. Published 2024 Jul 1. doi:10.1039/d3fo04687g

[3]Huang J, Han J, Rozi R, et al. Association between lipid accumulation products and osteoarthritis among adults in the United States: A cross-sectional study, NHANES 2017-2020. Prev Med. 2024;180:107861. doi:10.1016/j.ypmed.2024.107861

[4] Liu Y, Song F, Liu M, et al. Association between omega-3 polyunsaturated fatty acids and osteoarthritis: results from the NHANES 2003-2016 and Mendelian randomization study. Lipids Health Dis. 2024;23(1):147. Published 2024 May 17. doi:10.1186/s12944-024-02139-4

[5] Xue H, Zhang L, Xu J, et al. Association of the visceral fat metabolic score with osteoarthritis risk: a cross-sectional study from NHANES 2009-2018. BMC Public Health. 2024;24(1):2269. Published 2024 Aug 21. doi:10.1186/s12889-024-19722-0

2. By a methodological point of view Authors should clarify if they analyzed data stored in National Center for Health Statistics database or received samples on which they measured s-klotho.

Response (this part of the revision is highlighted in Green)：

Thank you for your kind reminder. We have clarified in the Methods section that the S-Klotho data for this study were obtained from the official NHANES website, rather than measured after receiving the samples.(line 155-156)

3. I think that is inappropriate define “races” instead of populations (table 1) Is very particular the classification in Non-Hispanic Black, Non-Hispanic White, Mexican American and other. Why authors do not used classical Caucasian, black, Hispanic and Asiatic definition?

Response：

We sincerely thank you for carefully reading. On the NHANES official website, participants are categorized into five racial groups: “Non-Hispanic Black,” “Non-Hispanic White,” “Mexican American,” “Other Hispanic,” and “Other Race”. Due to the relatively small sample sizes of “Other Hispanic” and “Other Race,” these groups have been combined for analysis, which is consistent with previous NHANES-related studies[6-8]. Based on this practice, our study categorized participants into “Non-Hispanic Black,” “Non-Hispanic White,” “Mexican American,” and “Other.”

References：

[6] Chu MT, Fenelon A, Adamkiewicz G, Zota AR. Federal Housing Assistance and Blood Lead Levels in a Nationally Representative US Sample Age 6 and Older: NHANES, 1999-2018. Environ Health Perspect. 2024;132(3):37004. doi:10.1289/EHP12645

[7] Zhang X, Wang X, Wang M, Qu H, Hu B, Li Q. Prevalence, awareness, and treatment of isolated diastolic hypertension in the United States. Eur J Intern Med. 2023;112:93-99. doi:10.1016/j.ejim.2023.03.026

[8] Zhou Y, Xu P, Qin S, Zhu Y, Gu K. The associations between dietary flavonoid intake and the prevalence of diabetes mellitus: Data from the National Health and Nutrition Examination Survey 2007-2010 and 2017-2018. Front Endocrinol (Lausanne). 2023;14:1250410. Published 2023 Aug 19. doi:10.3389/fendo.2023.1250410

4. Chu MT, Fenelon A, Adamkiewicz G, Zota AR. Federal Housing Assistance and Blood Lead Levels in a Nationally Representative US Sample Age 6 and Older: NHANES, 1999-2018. Environ Health Perspect. 2024;132(3):37004. doi:10.1289/EHP126454.Regarding ethics authors should clarify if participants in NHANES whose data were analyzed gave consent for their samples to be used in future research.

Response(this part of the revision is highlighted in Blue)：

Thank you for your kind reminder. We have revised the informed consent section for participants in the Methods section in accordance with your suggestion.(line 110-111)

5. I agree, as reported by the authors that “the conclusion may not be appropriate for generalization to other countries” than USA. In this view I would like to suggest to the authors to recruit a little cohort of OA patients (>100<300 individuals) on which evaluate the negative correlation between s-klotho and OA presence.

Response(this part of the revision is highlighted in Red)：

We sincerely appreciate your constructive suggestions. In response to your recommendations, we collected blood samples from 107 patients who were either hospitalized or attending outpatient clinics at the Orthopedic and Joint Surgery Department of the First Affiliated Hospital of Guangxi Medical University between August 6, 2024, and September 13, 2024. This group included 51 patients with osteoarthritis, diagnosed according to the 2018 Chinese Osteoarthritis Diagnosis and Treatment Guidelines[9], and 56 non-OA patients who did not meet the diagnostic criteria for osteoarthritis. We measured serum Klotho levels using an ELISA kit. The serum Klotho levels of the two groups were compared using independent samples t-tests, and the association between serum Klotho levels and osteoarthritis was analyzed using univariate logistic regression. The results indicated that serum Klotho levels were significantly higher in osteoarthritis patients compared to those without osteoarthritis (P<0.001). Each unit increase in serum Klotho levels was associated with a 23% reduction in the risk of osteoarthritis (OR: 0.77, 95% CI: 0.66, 0.90, P<0.001). These findings support the original results of our study. Although our sample size is currently limited and the information may be incomplete, our research preliminarily demonstrates a negative correlation between serum Klotho levels and the prevalence of osteoarthritis. This provides a theoretical basis for future large-scale, prospective cohort studies in different regions and populations to further explore the relationship between serum Klotho levels and osteoarthritis. The relevant section of the manuscript has been updated to reflect these results.(line 27-29, 40-42, 113-117, 127-131, 142-147, 158-161, 219-222, 277-286, 292-294, 340-341, 372-374, 380-385, 396)

References：

[9]Joint Surgery Group SoO, Chinese Medical Association. Guidelines for Diagnosis and Treatment of Osteoarthritis (2018 edition). Chinese Journal of Orthopaedics. 2018;38(12):705-15.

6. Discussion on the biological bases of klotho levels influencing OA presence might be enriched discussing data in the light of the following paper: Che QC, Jia Q, Zhang XY, Sun SN, Zhang XJ, Shu Q. A prospective study of the association between serum klotho and mortality among adults with rheumatoid arthritis in the USA. Arthritis Res Ther. 2023;25(1):149. doi:10.1186/s13075-023-03137-0

Response(this part of the revision is highlighted in Purple)：

Thank you for your valuable suggestion. We appreciate the perspective provided by the research of Che QC et al., which contributes to our understanding of Klotho's role in arthritis by highlighting its predictive value for mortality risk in rheumatoid arthritis patients. In response, we have included a description of the results from this study in the discussion section. This addition underscores the need for further investigation into the relationship between Klotho and mortality in osteoarthritis patients, which will be a focus of our future research. Thank you once again for your insightful suggestion.(line 331-335)

We tried our best to improve the manuscript and made some changes in the manuscript. 

These changes will not influence the content, conclusion and framework of the paper. We 

appreciate for editors/reviewers’ warm work earnestly, and hope that the correction will 

meet with approval. 

Once again, thank you very much for your comments and suggestions. 

Yours sincerely, 

Yue Qiu 

Corresponding author: 

Jia Li 

E-mail: jialee2005@126.com

---

## [Decision Letter · Decision Letter 1]

9 Oct 2024

Association between serum Klotho and the prevalence of osteoarthritis: a cross-sectional study from NHANES 2007-2016

PONE-D-24-19425R1

Dear Dr. Jia Li,

We’re pleased to inform you that your manuscript has been judged scientifically suitable for publication and will be formally accepted for publication once it meets all outstanding technical requirements.

Kind regards,

Calogero Caruso, MD

Academic Editor

PLOS ONE

Additional Editor Comments (optional):

Reviewers' comments:

Reviewer's Responses to Questions

**Comments to the Author**

1. If the authors have adequately addressed your comments raised in a previous round of review and you feel that this manuscript is now acceptable for publication, you may indicate that here to bypass the “Comments to the Author” section, enter your conflict of interest statement in the “Confidential to Editor” section, and submit your "Accept" recommendation.

Reviewer #1: All comments have been addressed

2. Is the manuscript technically sound, and do the data support the conclusions?

Reviewer #1: Yes

3. Has the statistical analysis been performed appropriately and rigorously? 

Reviewer #1: Yes

4. Have the authors made all data underlying the findings in their manuscript fully available?

Reviewer #1: Yes

5. Is the manuscript presented in an intelligible fashion and written in standard English?

Reviewer #1: Yes

6. Review Comments to the Author

Reviewer #1: Authors have significantly improved the manuscript. Their answers have completely satisfied the questions, criticisms and suggestions. In my opinion paper might be fully recommended for publication on Plos One Journal.

7. PLOS authors have the option to publish the peer review history of their article (what does this mean?). If published, this will include your full peer review and any attached files.

Reviewer #1: No

---

## [Editor Report · Acceptance letter]

11 Oct 2024

PONE-D-24-19425R1 

PLOS ONE

Dear Dr. Li, 

I'm pleased to inform you that your manuscript has been deemed suitable for publication in PLOS ONE. Congratulations! Your manuscript is now being handed over to our production team.

Kind regards, 

on behalf of

Prof. Calogero Caruso 

Academic Editor

PLOS ONE